# Social and Emotional Skills in at-Risk Adolescents through Participation in Sports

**DOI:** 10.3390/sports12070181

**Published:** 2024-06-27

**Authors:** Andreia P. Teques, Rita F. de Oliveira, Michala Bednarikova, Maurizio Bertollo, Grzegorz Botwina, Anastasiya Khomutova, Hamit Emir Turam, İlknur Dinç, Marcos López-Flores, Pedro Teques

**Affiliations:** 1European Network for Innovation and Knowledge (EUNIK), 3815 JA Amersfoort, The Netherlands; apereirateques@ipmaia.pt (A.P.T.); grzegorz.botwina@eunik.org (G.B.); marcos.lopez@eunik.org (M.L.-F.); 2N2i, Polytechnic Institute of Maia, 4475-690 Maia, Portugal; 3School of Applied Sciences, London South Bank University (LSBU), London SE1 0AA, UK; r.oliveira@lsbu.ac.uk; 4European Federation of Sport Psychology (FEPSAC), B-1000 Brussels, Belgium; bednarikova@sportpsych.sk (M.B.); m.bertollo@unich.it (M.B.); a.khomutova@brighton.ac.uk (A.K.); 5Department of Medicine and Aging Sciences, “G. d’Annunzio” University of Chieti-Pescara, 66100 Chieti, Italy; 6Faculty of Management, University of Warsaw, 03-772 Warsaw, Poland; 7School of Sport and Health Sciences, University of Brighton, Brighton BN20 7SP, UK; 8Istanbul Culture and Sport Association, 34846 Istanbul, Turkey; emir.turam@gmail.com (H.E.T.); ilknur@demturkey.com (İ.D.); 9Research Centre in Sport Sciences, Health Sciences, and Human Development (CIDESD), 5001-801 Vila Real, Portugal

**Keywords:** at-risk youth, conduct problems, emotional skills, institution, social support

## Abstract

(1) Background: Adolescents who are under the care of child and youth institutions are vulnerable due to factors that can include disruption to family structure or education and adverse experiences. They often experience poor or unstable support systems, leaving them at risk of delinquency. In this context, sports engagement may provide a stable structure and have positive effects in this population. Therefore, the purpose of this study was to explore the perceptions, knowledge, beliefs, and behaviors regarding social and emotional skills among at-risk adolescents, with a specific focus on their engagement in sports. (2) Methods: Ninety-six adolescents aged 12 to 17 years (66 female, 30 male), residing in child and youth care institutions across Italy, Portugal, the UK, and Turkey, participated in this study. The participants were divided into 14 focus groups, each with six to eight participants of similar ages. (3) Results: Thematic analysis revealed four main themes: Emotional causes of behavioral problems; Emotional skills to regain control; Social support makes sports worthwhile; Sport as a socio-emotional resource. (4) Conclusions: The study findings highlight that although sports create many challenging emotional situations for these adolescents, they also provides resources that may help them cope with their emotions within and beyond sports.

## 1. Introduction

As part of normal development, adolescents seek novel experiences to improve their skills, satisfy their curiosity, provide leisure, or foster new connections [1]. Adolescents who have had disruption to their family structure or education, or had adverse experiences, sometimes go further and engage in risky behaviors (e.g., substance use, brawling, crime); therefore, they are considered to be at-risk adolescents [2]. Moreover, adolescents with difficulties in emotional regulation are more likely to engage in offenses due to their limited ability to anticipate and consider the potential consequences of their actions [3]. This can be made worse by impulsive actions, influenced by social pressures, which may lead to adverse and unlawful outcomes [4]. At-risk adolescents may be taken under the care of institutions for the purpose of safeguarding them.

Sports engagement often provides structure and social support, which are helpful to adolescents in general and may be particularly helpful for adolescents at risk [5]. However, access to sports participation is not always available in families facing challenging socio-economic conditions due to their associated costs and other factors [6,7,8]. Difficulties in interpersonal relationships within the family also impact access to sports, and adolescents experiencing poor emotional bonds with their families may seek sensations and stimuli in leisure activities that deviate from societal norms [9]. One study sought to evaluate the effects of a sports-based intervention in preventing delinquency in at-risk adolescents in a well-controlled study [10]. Although they found small positive effects, the mechanisms for these effects were unclear because no effects were found on the risk or protective factors assessed.

Significant figures can play an important role in guiding the development of adolescents at risk by providing personal, affective, and social support, thereby promoting adaptive growth [11]. The relational experiences with positive teachers or coaches [12] may foster the development of more positive self-perceptions [3] and emotional regulation. On the other hand, the peer group may also provide nurturing connections [13,14]. The acceptance and sense of belonging within the peer group can significantly predict social and emotional competencies, such as self-esteem [15] or a sense of personal identity [16]. Peer relationships become more prevalent and significant during adolescence, leading to improved social and emotional competence [17,18]. Thus, the sports context provides an environment where positive coaching and positive peer support are reported to contribute to positive emotional regulation [19,20,21]. However, the sports context can also elicit strong emotions, which may be challenging for at-risk adolescents to regulate. It is therefore unclear what benefits and challenges sports may bring to the emotional regulation of adolescents at risk.

Despite research to date on the social and emotional competencies of at-risk adolescents having developed knowledge about their risky behaviors [22], emotional regulation difficulties [23], limitations on sports participation due to families’ financial constraints [24], and the importance of significant others in promoting emotional and social competencies in these adolescents [25], there is still a need to explore the emotional factors associated with sports participation (i.e., what are the factors that trigger emotions in these adolescents?), mechanisms of emotional regulation (i.e., how do these adolescents regulate their emotions?), and the potential emotional and social benefits of sports participation for these adolescents (i.e., what are the emotional and social consequences of sports participation for these adolescents?).

Adolescents experiencing disruptions in their family structure may engage in risky behaviors [26], potentially exacerbated by emotional regulation difficulties. In turn, these behaviors can lead to substance abuse, violence, and theft [27]. Therefore, understanding the nuances of the connection between the context of sports participation and the social and emotional competencies of at-risk adolescents can be helpful in ensuring an interpretative framework for this population. Hence, the purpose of this study was to explore the perceptions, knowledge, beliefs, and behaviors regarding social and emotional skills among at-risk adolescents, with a specific focus on their engagement in sports.

## 2. Materials and Methods

### 2.1. Participants

Ninety-six adolescents aged 12 to 17 years (66 female, 30 male), residing in child and youth care institutions across Italy, Ireland, Portugal, the UK, and Turkey, participated in this study (please see Table 1 for more information). The diversity of perspectives and institutional care practices across different social systems can help understand the cultural processes and common psychological challenges faced by institutionalized adolescents [28]. On average, these adolescents had been under the care of an institution for 9 months (ranging from 1 to 48 months). The participants were divided into 14 focus groups, each with 6–8 participants of similar ages (i.e., 12–14 or 15–17). This age-based grouping aimed to facilitate communication and interpretation of the discussions [22].

The child and youth care institutions were contacted and informed about the goal of the study and granted permission for the adolescents’ participation. The institutions were responsible for selecting and inviting the adolescents who met the inclusion criteria: (1) adolescents aged 12 to 17, (2) who were, at the time of the study, dependent on an institution for direct care, and (3) who practiced sports regularly. Before data collection, the researchers informed the adolescents of the study goals, purpose of their participation, data protection regarding confidentiality and anonymity, and their right to withdrawal. The study received ethical approval from the Ethics Committee of the Research Centre of the Polytechnic Institute of Maia (Ref. 002/06/2021).

### 2.2. Data Collection

The interview guide used in the focus groups was developed according to interpretive research methods and contained 14 open-ended non-leading questions [29]. To favor participation [30] the initial question was open-ended: “Tell us more about the sport that you do, and what you like about it?”. The interviews then followed a stepwise approach of three levels of questions from general to specific questions [31], following a timeline from the past, present, and future [32]. Based on the literature about psychosocial challenges in adolescents at risk, e.g., [33,34], the interview guide included topics related to emotional recognition (i.e., “Can you recognise and name emotions that you feel?”), emotional regulation (i.e., “How do you manage your emotions during your sports practice?”), impulse control (i.e., “What behaviours of other athletes make you react emotionally?”), the role of physical activity in life (i.e., “How can sport help you in your future life?”), the role of physical activity in emotional regulation (i.e., “How can physical activity help you to overcome negative feelings?”), or coping strategies they may use to deal with adversity (i.e., What other strategies do you know to overcome less positive feelings?). The focus groups took place on the premises of the social institutions and were recorded in audio for later analysis of the content, after obtaining permission from all participants. The discussions were managed by two researchers. One led the facilitation of the discussion process and ensured that all participants had the opportunity to express their opinions, while the other ensured the proper audio conditions and kept notes to complement the understanding of the different perspectives and experiences of the participants. This procedure was carried out with both age groups. The focus groups took about 45 min.

### 2.3. Data Analysis

The audio content was transcribed in full into digital format (resulting in 115 single-space pages) and then analyzed using thematic analysis [35], which included familiarizing with the data, defining initial codes, classifying the codes into themes, reviewing and refining the themes, and finally, writing up the analysis. Nvivo 12 was used to facilitate data analysis by identifying passages with keywords, extracting coded passages, and synthesizing them to identify themes and sub-themes. The process followed an inductive approach, in which themes were identified from the data collected. Although the data were analyzed inductively, the researchers did not code the data in an epistemological vacuum and the data analysis was carried out through both inductive and deductive processes [36]. The international nature of this research and logistical issues concerning access to the population meant that the number of focus groups was determined a priori and that all data were analyzed (instead of stopping analysis at data saturation).

To lend credibility to the data analysis, the first author discussed their interpretations with another co-author who had the role of ‘critical friend’ in a first instance [37]. After the first author had finalized the initial analysis, they shared a list of themes, sub-themes, and example quotes with the co-author for discussion to ensure that the themes accurately represented the data. Additionally, the analysis was discussed by the research team members, through subsequent discussions about the themes and their contextualization within the theoretical model [38]. These discussions enhanced the refinement of the themes and continued until the writing of the scientific report. In addition, they helped to enhance rigor and reflexivity by exploring different interpretations of the data and gain access to potentially unobservable blind spots [39]. The interpretive patterns of the data under analysis enabled the addition of new knowledge in this area of study [40].

## 3. Results

There were four main themes, each with several sub-themes. They were (1) Emotional causes of behavioral problems, which included: Win-lose situations; Managing externalizing behaviors; Dealing with mistakes; Opponent behavior; (2) Emotional skills to regain control, which included: Emotional awareness; Affective enhancement; Emotional understanding; Emotional regulation; (3) Social support makes sports worthwhile, which included: Helping others; Being helped; Empathy for teammates; and (4) Sports as a socio-emotional resource, which included: Well-being; Emotional regulation; Values for life; Positive feelings. Figure 1 represents an inductive model built upon the sub-themes and their interrelationships.

### 3.1. Emotional Causes of Behavioral Problems

Participants reported protective and exacerbating factors associated with behavioural problems, such as Win–lose situations in sports, Opponent behaviors, Managing externalizing behaviors, and Dealing with mistakes. Behavioral problems include hostility, breaching the law, and destructiveness. Adolescents that engage in these behaviors may experience a range of unfavorable consequences, such as academic failure or substance misuse. Behavioral issues can also show up in sports, where they take the form of aggression, dishonesty, and other types of rule breaking.

#### 3.1.1. Win–Lose Situations

Win-lose situations in sports manifested as the personal challenge in grappling with unfulfilled expectations. For example, one participant stated that “... in football I get angry when we lose a game, and when they’re trying to stare at me and that, and I lose the ball or when they get a foul and it’s hardly a foul” (PT3). Encountering a loss in sports can lead to psychological impacts on participants, inducing negative emotions like pessimism, diminished enthusiasm, or uncertainty: “After our team lost the final game, I couldn’t shake off the feeling of disappointment. It was like a punch to the gut, knowing we came so close but fell short in the end. Then I lose control” (TK6).

#### 3.1.2. Managing Externalizing Behaviors

Managing externalizing behaviors was made easier in the context of sports because sports involve clear rules. The following quote exemplifies this aspect: “What separates me from football are the rules because I can’t go there and take a punch because you took the ball away from me. I can’t do that, so we always have to be careful” (PT2). This underscores the potential efficacy of sports as a mechanism for emotional self-regulation and for the successful reintegration of at-risk youth into societal frameworks.

#### 3.1.3. Dealing with Mistakes

Dealing with mistakes was seen from diverse perspectives given different values or goals expressed by the adolescents. For instance, a UK adolescent expressed the following on this matter: “One of the things that make me feel negative is playing badly in the match or doing something wrong” (UK2). However, one Italian adolescent had a very different view on mistakes: “Messing up isn’t just about, like, making mistakes during the game. It’s about how we think about winning, you know? When all that matters is winning, any little mistake feels like a huge fail. But if you are all about putting in the work, mistakes are just chances to get better, you know what I mean?” (IT5). These quotes illustrate the diversity of attitudes regarding how adolescents deal with mistakes and highlight potential cultural differences in interpreting success and failure in sports practice.

#### 3.1.4. Opponent Behavior

Although regulations outline behaviors that could lead to disciplinary action, there is often an occurrence of inappropriate attitudes and aggressive conduct that goes beyond ethical boundaries. The participants in this study reported that they felt inclined to react angrily in the face of cheating or ethically questionable behaviors: “… I get angry when they try to cheat, pretend to hurt or something.” (PT5). Aggressiveness is demonstrated through behaviors—whether physical, verbal, or gestural—adopted by the adolescent directed at another, with the intent to cause harm or damage. In the present study, the aggressor frequently perceives the potential harm inflicted on the opponent as collateral, deeming it legitimate and justifiable: “Sometimes in the heat of the game, you just go all out. Like, it’s not personal, but if I get physical or get in someone’s face to win, I‘ll do it. It’s all part of the game. If the other guy gets hurt, well, that’s just how it goes sometimes” (PT15).

### 3.2. Emotional Skills to Regain Control

These adolescents at risk are often exposed to high levels of stress (e.g., socioeconomic disadvantage, peer pressures, traumatic experiences) and emotional skills can help them to manage their stress and avoid engaging in harmful coping mechanisms, such as substance abuse or violence [41]. Some of the emotional skills that emerged from the content were related to Emotional awareness, Affective enhancement, Emotional understanding, and Emotional regulation.

#### 3.2.1. Emotional Awareness

Through participants’ words, a common theme emerged: a profound sense of frustration with their own emotional volatility and a desire to regain control over their reactions: “I am a very angry person, but I don’t want to get angry. For example, in some things, even in the smallest thing, I can upset the other person very quickly, it makes me very angry. For example, I can put your hand on my shoulder, put your hand on my shoulder and push your hand away” (TK6). This shared experience underscores the depth of emotional confusion experienced by these adolescents and the challenges they face in maintaining stable relationships. These adolescents grapple with the daily struggle of reconciling their inner emotional experiences with their social relationships.

#### 3.2.2. Affective Enhancement

Adolescents expressed how emotions and thoughts sometimes intertwine in their mind during sports practice. During the focus groups, it was possible to recognize how they try to figure out whether they are feeling happy or not and what is happening during their sport experiences. One adolescent stated, “Am I happy, or am I unhappy, what is my current situation? Most of the time I can identify it, but again most of the time we can’t identify it when something happens. But if I know myself, I can say that if I am doing this, I have done it, but if I cannot do it, I cannot say that I cannot do it. Because I question myself about what I did wrong” (PT12). Despite uncertainty, they are determined to understand themselves better and learn from their experiences. It is as if they are on a journey to discover who they are and how they can grow from what they go through: “It’s good to put yourself out of your comfort zone ‘cause that’s where growth is. Especially if you’ve not got something like with a specific goal that boxing is. I could see myself in life just staying in the same place really easily. I don’t know. I think it [boxing] builds a lot of character” (UK4).

#### 3.2.3. Emotional Understanding

Participants provided valuable insight into how emotions can impact their behavior, especially during sports. They understood that when they cannot control the strong feelings bubbling up inside, whether it is overwhelming sadness or intense anger, they tend to make more mistakes and have more problems with others in sports practice: “I can’t control the feeling inside of me. It comes out as either too much sadness or too much anger. That’s why I can have sudden explosions, I can make some of my friends upset, but not really on purpose” (IT2). The outburst of feelings catches the adolescent off guard, provoking intense reactions even if they never meant to upset their friends or teammates. It is a tough spot to be in, but understanding and learning to deal with these emotions is all part of growing up. It seems that they are trying to understand how to handle their feelings while enjoying sports.

#### 3.2.4. Emotional Regulation

The participants reported their approach to how they regulate emotions. An adolescent from the UK shared that “If there is an anger threshold or something, it would be better if we get away from that environment before the anger comes” (UK6). Often, they demonstrated that they are aware of their emotions and trying to handle them in a clever way. While this kind of self-awareness may be an important aspect for their adaptation, they may also represent an ongoing struggle in dealing with their feelings: “Yeah, when I play, I‘m always aware of how I‘m feeling. It’s like I‘m constantly trying to manage my emotions, you know? It’s important for me to stay focused and in control, but it’s not always easy. Sometimes, I feel like I‘m battling my own emotions on the field, but I know I‘ve got to keep pushing through” (TK9).

### 3.3. Social Support Is What Makes Sports Worthwhile

Social support emerged as a particularly important theme for the adolescents when they used expressions like, “working together” or “being available to listen to teammates’ problems”. They reported the importance of Helping others, Being helped, and the Empathy for teammates.

#### 3.3.1. Helping Others

This sub-theme emerged in the context of peer motivation and support, and in enhancing group cohesion. Adolescents indicated their commitment to motivating their teammates reflecting an understanding of the interpersonal dynamics inherent in team sports, wherein mutual encouragement and support foster a sense of camaraderie and collective achievement: “My constant focus is on motivating my teammates” (IT8). Participants’ narratives shed light on the multifaceted nature of social interactions and support mechanisms inherent to adolescents at risk in sporting environments, revealing the interplay between individual experiences and collective dynamics: “We may have issues with each other, but when it comes to defending our group, we’re here! Here [at the care institution], we need each other. And when someone needs help or needs to talk, we try to support each other” (TK11). Despite occasional discord, they show solidarity in defending their collective because they feel that inside the institution, mutual reliance is paramount. Whenever an individual requires assistance, they unite to offer support and empathy, fostering a sense of unity.

#### 3.3.2. Being Helped

The adolescents explored who were the individuals who they would trust the most to help them. They identified those with whom they are willing to share important information about their lives and personal problems. Some adolescents defined the characteristics of people they trusted: “(…) so, first, I think, you have to be a woman. Then, to have empathic capabilities, a certain emotional intelligence, a maturity to understand and know how to help with communication skills, to know what to say” (PT1). Although they saw themselves as someone who would help their peers, they did not immediately identify their peers as the ones to turn to for help.

#### 3.3.3. Empathy for Teammates

This aspect played a pivotal role in cultivating positive dynamics among adolescents at risk in sports. For instance, an Italian adolescent expressed, “I think I [show] respect. If someone in the team tells me to stop doing something, I‘m not going to stand there doing what they tell me not to do” (IT3). This underscores the significance of mutual respect and consideration among teammates. Sports offer a nurturing environment for adolescents confronting challenges, where empathy and understanding are vital. By showing regard for their teammates’ boundaries and needs, these adolescents foster a culture of fraternity. This empathetic approach may fortify interpersonal bonds, enhance collaboration, and show a concern for acknowledging others’ perspectives.

### 3.4. Sports Is a Socio-Emotional Resource

Participants expressed that sports provide a safe and structured environment where they can learn and grow. It also helped adolescents develop coping mechanisms for dealing with adversity and stress. Regarding this theme, adolescents emphasised Sport as a resource to promote well-being, Sports as a source of positive feelings, Sport as a resource to regulate emotions and Sport as a resource to transfer values to life.

#### 3.4.1. Sport as a Resource to Promote Well-Being

A sub-theme that emerged from the data was related to perceptions about the positive effects of sports, highlighting the health-related benefits. Adolescents saw sports engagement as a self-care activity: “It makes us take care of ourselves. For example, almost all of Turkey is currently dealing with obesity and it protects us from obesity in the future. There is no such risk in the future; we become healthy” (TK8). This awareness indicates an understanding of these adolescents about how leading an active lifestyle impacts overall health and future well-being. It also reflects an appreciation for health as a crucial aspect of personal development, highlighting a narrative of self-care and individual responsibility: “For me being active isn’t just about winning games. It’s about taking care of myself and realizing that my health matters. When I‘m out there on the field, I‘m not just playing for the team. I‘m playing for my own well-being too. It’s like I‘m investing in my future” (IT13).

#### 3.4.2. Sports as a Source of Positive Feelings

Actions and emotions in sport occur within a situational context (i.e., involving the interaction of individuals, tasks, and the environment). For these adolescents, sports are an opportunity to feel and express positive emotions. In fact, positive feelings about sport emerge as a protective factor against conduct problems: “… the positive part is when I learn the moves well, everything calmly and safely” (TK8). This sentiment seems to underscore the pivotal role of positive emotional experiences in fostering a sense of mastery, security, and overall well-being within the sports domain.

#### 3.4.3. Sport as a Resource to Regulate Emotions

Adolescents saw sports as a constructive way for them to deal with external challenges through diversion and catharsis. Through sports, adolescents at risk could channel their emotions and energy, affording them a respite from external stressors and a platform to discharge frustrations: “I mean, you fight outside, there will be problems with teachers at school or with others out of the school, so you can come here and distract yourself here, you can forget what happened there and discharge here” (UK2). They recognize the therapeutic value of sports in their lives, offering a healthy outlet for managing stress and navigating interpersonal conflicts: “it’s [sports is] like our sanctuary, helping us cope with life’s challenges and protecting us from external pressures” (PT11). This emphasizes the potential of sports to act as a protective buffer against adversities felt in other areas of life, including academic and social.

#### 3.4.4. Sport as a Resource to Transfer Values for Life

The perspective of sports as a vehicle for instilling values in the lives of these adolescents emerged from the text. For example, one participant said, “we learn discipline by doing sports, coming to training, etc. I think this will contribute positively to our business life or basketball life in the future” (TK6). The adolescents understood that the values learned through sports engagement applied to their lives both inside and outside sports, both at present and in the future. Sports engagement was particularly linked with values such as discipline, commitment, and effort: “I understand that putting in the work now can help me succeed later, whether it’s in my dream job or staying on the team”. This highlights the transferable skills cultivated through sports that extend beyond sports to enrich various facets of their lives. It is noteworthy that the younger adolescents believed they could become professional athletes and saw the value of sports more narrowly, while older adolescents expressed how sports instilled values which they viewed as essential for success both within and beyond sports.

## 4. Discussion

The purpose of this study was to explore the perceptions, knowledge, beliefs, and behaviors regarding social and emotional skills among at-risk adolescents, with a specific focus on their engagement in sports. The main themes were Emotional causes of behavioral problems, Emotional skills to regain control, Social support makes sports worthwhile, and Sports as a socio-emotional resource. These main themes were found in focus groups across five European countries, and therefore, an important contribution of this study is to show commonalities in the influence sport can have in the emotional development of adolescents at risk. The study findings highlight that although sports create many challenging emotional situations for these adolescents, they also provide resources that may help them cope with their emotions within and beyond sports.

Adolescents shared their difficulties in managing social interactions, particularly in the context of win–lose situations, in dealing with their own mistakes and with opponents. Although adolescence is a time when all adolescents learn to manage social interactions, at-risk adolescents may find it more challenging to deal with their own anger and aggressiveness. This is corroborated in the literature, e.g., [42,43], taking into account that the quality of relationships and dynamics developed in childhood and youth is a determinant for the promotion of social and emotional competences. To the extent that it is reflected in the ability to perceive their context in the social interactions they establish, children who develop in safe contexts that meet their basic needs have greater interpersonal and affective skills when compared to children who experience insecurity, neglect, or abuse [3]. Institutionalized adolescents are often perceived as aggressive, with fewer conflict resolution competences, fewer pro-social skills, and with more behavioral problems, resulting in poorer and less reciprocal peer social support networks, e.g., [44,45,46]. These recent studies show that the interaction difficulties with caregivers, teachers, and peer groups can lead to their further exclusion. Therefore, it is likely that the social difficulties that the adolescents in this study reported in the sports context are also visible in other contexts, such as school and the care institution.

The data from the present study indicate significant challenges in emotional evaluation and expression. The data from the present study point to significant hurdles in emotional learning, including emotional evaluation and expression, underscoring the adolescents’ struggles in acquiring the skills needed for self-regulation and effectively managing interpersonal conflicts. Such difficulties can hinder, for instance, the processing of social information or the interpretation of facial expressions depicting emotions [3]. Institutionalized adolescents may also exhibit emotional suppression as a non-adaptive method of managing emotions, which is negatively associated with mental health [47]. However, the context of sports offers an environment where certain levels of anger and aggression are permissible, potentially affording adolescents the opportunity to develop some skills to manage and channel their emotions more effectively. For instance, a recent review found that young participants with behavioral and violence problems showed a positive response to engaging in martial arts [48]. The adolescents in our study demonstrated a clear desire to regain control over their emotions, acknowledging the role of sports rules in tempering their emotional outbursts to some extent. However, they also acknowledged the inherent challenge in doing so.

One protective or mediating factor of well-being in at-risk adolescents is their perception of current social support, because it fosters a sense of belonging and integration within the community [15]. Here, the care institution can be seen as a relational space where change occurs through the relearning of social relations within the group. Keil et al. [49] refer to this social phenomenon as the development of “hyper-cooperativeness”, facilitated by cohabitation and group participation in sports or other recreational activities. Sociability becomes a protective factor for temperament characteristics such as emotional regulation and negative lability [50]. The adolescents in our study referred to the positive impact of social support from teammates, and used sports as a means to regulate their emotions, emphasizing the relief from daily emotional tension, as well as the benefits for their physical and psychological well-being. Emotional regulation refers to the ability to influence one’s emotional state, considering the significance of emotions experienced and expressed in various situations and contexts [51]. Within the context of collective care and cohabitation, interactions among children play a pivotal role in enhancing empathic and compassionate abilities, as well as in forming connections within and outside the care institution from a social perspective [52]. Moreover, in the context of sports, the practice of regulation is constant, as athletes encounter diverse emotions, and their regulation directly impacts performance, personal well-being, and positive social relationships. This experiential knowledge gained through sports practice underscores the intertwining of emotions and the social outcomes of sports [12].

In addition to emotional management in sport, there is also the factor that fosters the creation of new values, principles, and life goals expressed by the adolescents in this study, such as responsibility, commitment, discipline, and respect, not only as a result of the relationships established with teammates, but also by the presence of coaches and the rules of the sport. Coaches may be seen as models of emotional regulation who value them and integrate them into the community. According to Braun and Tamminen [12], emotional skills are related to the interpersonal regulation of the coach’s emotions, which influences the emotional regulation techniques of the athletes themselves. It is unquestionable that institutional practices have an impact, for example, through programs to promote socio-emotional literacy and empowerment and physical and psychological health [53], but sports may be a precious context where adolescents at risk have the opportunity to learn to regulate their emotions under the guidance of a coach, surrounded by social peer support, with respect for the sports rules, and perhaps for the love of a game.

### 4.1. Limitations and Further Research

Even though this study has produced several insightful reflections, there are a few limitations to take into account as well as possible directions for further research. Participants consisted only of adolescents who needed institutional care and who were already involved in sports. As a result, it is unclear if the results would also apply to adolescents less inclined to engage in sports. Future studies may be able to use control participants in cross-over designs to assess the effects of sports engagement on at-risk adolescents. In a study including early adolescents, Fine et al. [54] noted variations in emotional and social components based on the geographical context, which were not evident in the present study of four countries. Future investigations might incorporate adolescents from more diverse socioeconomic backgrounds, geographic locations, and cultural contexts.

Also, the study relied on focus groups as the primary data collection method. Although focus groups offer nuanced insights into participant experiences, they may introduce biases such as social desirability or conformity effects [55]. To mitigate these issues, future research can employ mixed-methods approaches, integrating qualitative data from focus groups with quantitative measures to explore and confirm findings. This integrative approach not only strengthens the validity of research findings but can also offer better insights for designing targeted interventions and support programs tailored to the needs of this population.

Future research should use these and other results, e.g., [10,56], to design and examine the efficacy of sports-based interventions targeting emotional competencies among at-risk adolescents. In line with longitudinal findings that showed that higher participation in sport is related with improvements in perceptions of social competence during childhood to adolescence [57], future research could track at-risk adolescents’ progress over time to provide insights into the enduring impact of sports participation on social and emotional development, as well as associated outcomes like academic performance and psychological health. Additionally, experimental studies evaluating specific intervention strategies, such as mentoring programs, e.g., [58], could offer evidence-based guidance for designing effective sports programs for at-risk youth.

### 4.2. Practical Implications

Following from the results of this study, three guidelines for intervention practice and social pedagogy policies can be structured. First, despite the literature already highlighting emotional skills training programs in children and adolescents in sports [59], sports coaches and instructors can be instructed to develop practical approaches in training sessions aimed at strengthening emotional and social skills, such as emotional self-regulation or conflict resolution with at-risk adolescents. This type of intervention is already implemented with teachers in the educational context with some positive results [60]. Second, it seems relevant that coaches working with this population receive specific training on how to address the emotional and social needs of at-risk adolescents. This includes strategies to promote supportive and safe environments to develop interpersonal skills, for example using principles of non-linear pedagogy and positive sports coaching [8,19]. Finally, the development of partnerships between sports institutions and care institutions can facilitate at-risk adolescents’ access to structured sports programs and emotional support provided by specialists (e.g., psychologists and sports coaches trained in this area). These partnerships can enhance the sharing of human resources, specialized technical training, and the organization of joint events to promote the inclusion and well-being of at-risk adolescents.

## 5. Conclusions

In conclusion, the findings point to several factors that can influence the behaviors of at-risk adolescents in sports, namely difficulties in managing social interactions, particularly in the context of win–lose situations, and in dealing with their own mistakes and with opponents. Emotional skills such as awareness, appraisal, understanding, and regulation of emotions were identified as relevant factors that can promote well-being and develop values. Although sports create challenging emotional situations for adolescents at risk, they also provide resources that may help them cope with their emotions within and beyond sports.

## Figures and Tables

**Figure 1 sports-12-00181-f001:**
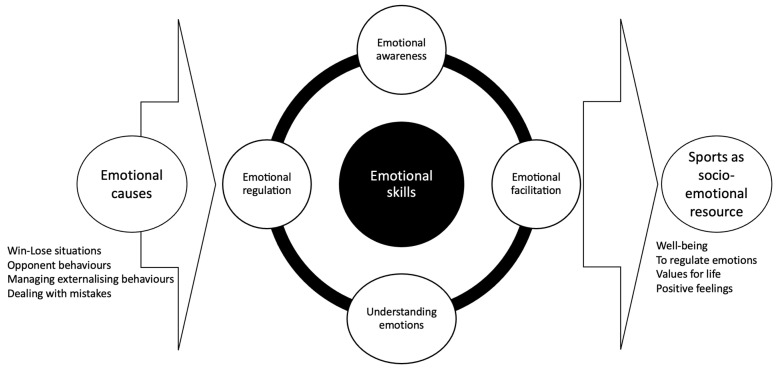
Inductive model about institutionalized adolescents’ perceptions, knowledge, beliefs, and behaviors of social and emotional skills through sports.

**Table 1 sports-12-00181-t001:** Demographic characteristics of study participants.

Characteristics	Italy	Ireland	Portugal	United Kingdom	Turkey	Total
Number of participants	16	16	32	16	16	96
Gender (Female/Male)	12/4	8/8	18/14	12/4	16/0	66/30
Ages 12–14	8	8	16	8	8	48
Ages 15–17	8	8	16	8	8	48

## Data Availability

Data are available from the corresponding author on request due to containing information that could compromise the confidentiality of research participants.

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
