# Peer review of "Social and Emotional Skills in at-Risk Adolescents through Participation in Sports"

_sports, 2024, doi:10.3390/sports12070181_

Round 1
Reviewer 1 Report
Comments and Suggestions for Authors
The authors need to address the following for the article to be considered for publication.
A substantial body of research has examined the development of social and emotional skills in at-risk adolescents who participate in sports. In the Introduction section, the authors need to justify the necessity of this study in light of the limitations of previous studies. In its current form, it is difficult to ascertain how this study contributes to the existing body of literature.
Moreover, if the authors are particularly interested in understanding emotional recognition, impulse control, the role of physical activity in life, and emotional regulation among at-risk adolescents who participate in sports, they need to describe in the Introduction section why these things are worthwhile to be investigated.
In the Method section, the authors need to present a table that outlines the characteristics of the study participants. Furthermore, the authors need to justify the combination of data from different countries.
In subheading 3.1.3, the authors discuss the diverse values of goals expressed by adolescents in different countries. However, by describing one or two adolescents’ transcripts from different countries, the authors provide an arbitrary and unscientific interpretation. As a qualitative study, the authors need to examine their responses in depth in order to gain insight into the reasons behind the differences in values and goals. Along the same lines, the current format of the manuscript in the Results section is inadequate for providing insight into the reasons and thoughts beyond the participants’ transcripts.
The content of lines 189 to 194 reflects the achievement motivation. The interpretation of the result can be enhanced by referencing Nicholls, J. G. (1984), Achievement motivation: Conceptions of ability, subjective experience, task choice, and performance. Psychological Review, 91(3), 328-346.
In the discussion, the authors need to demonstrate how this study makes a unique contribution to the existing literature. Additionally, they need to provide a clear and detailed account of the practical implications of the study.
Author Response
The authors need to address the following for the article to be considered for publication.
R: Thank you for the comments. We realize that our responses to the comments have improved the quality of the manuscript.
- A substantial body of research has examined the development of social and emotional skills in at-risk adolescents who participate in sports. In the Introduction section, the authors need to justify the necessity of this study in light of the limitations of previous studies. In its current form, it is difficult to ascertain how this study contributes to the existing body of literature. Moreover, if the authors are particularly interested in understanding emotional recognition, impulse control, the role of physical activity in life, and emotional regulation among at-risk adolescents who participate in sports, they need to describe in the Introduction section why these things are worthwhile to be investigated.
R: We have now extended the introduction to more clearly justify the need for this study in light of the previous literature and we have expanded the content that describes the need to study these aspects in this population and in the sports context. Please see paragraph between line 62 and line 77 and text between lines 78 and 83.
- In the Method section, the authors need to present a table that outlines the characteristics of the study participants. Furthermore, the authors need to justify the combination of data from different countries.
R: Table 1 was inserted to specify the characteristics of the participants (Please see Table 1). Additionally, the inclusion of participants from different countries was justified (Please see text between lines 90 and 93). The research consortium composed of researchers from different countries wanted to explore what the sports context can bring to adolescents at risk beyond national and institutional levels and for that reason it was beneficial to have the diversity in countries and institutions.
- In subheading 3.1.3, the authors discuss the diverse values of goals expressed by adolescents in different countries. However, by describing one or two adolescents’ transcripts from different countries, the authors provide an arbitrary and unscientific interpretation. As a qualitative study, the authors need to examine their responses in depth in order to gain insight into the reasons behind the differences in values and goals. Along the same lines, the current format of the manuscript in the Results section is inadequate for providing insight into the reasons and thoughts beyond the participants’ transcripts.
The content of lines 189 to 194 reflects the achievement motivation. The interpretation of the result can be enhanced by referencing Nicholls, J. G. (1984), Achievement motivation: Conceptions of ability, subjective experience, task choice, and performance. Psychological Review, 91(3), 328-346.
R: We have now removed the text and replaced it with a brief interpretation that is more closely aligned with the quotes and in line with the rationale of presenting the results (please see between lines 218 and 220).
- In the discussion, the authors need to demonstrate how this study makes a unique contribution to the existing literature. Additionally, they need to provide a clear and detailed account of the practical implications of the study.
R: The main themes were found in focus groups across five European countries and therefore an important contribution of this study is to show commonalities in the influence that sport can have in the emotional development of adolescents at risk. This is now more clearly stated in the discussion (please see lines 391 and 394). A paragraph on the "Practical implications" of the study has also been added (please see lines 492 and 508).
Reviewer 2 Report
Comments and Suggestions for Authors
I must point out that it was a pleasure to read this article. It is written very specifically, without expanding too much where it is not necessary (for example, in the introduction). Although qualitative research articles are often much larger in scope than quantitative research (and understandably so), in this particular case I preferred the specificity.
However, I would like to make comments that the authors of the article may consider.
The study included 14 focus groups, which is a lot in itself. On the other hand, despite such a large number of research participants and groups for a qualitative study, was it sufficient to obtain detailed information in order to response to the research purpose? I mean data saturation.
Did the research participants give permission to audio record the interview data?
Conducting a focus group is not an easy task for researchers. Therefore, it is recommended that more than one researcher conducts the focus group interview. I missed information about how the group discussion itself went. Also, what about brief reflections of the researcher himself. In addition to the audio recording, was there additional information captured?
Finally, research validity (trustworthiness). You focused on credibility (what is good). And what you have done in order to ensure credibility was based on this source - Holt, N., & Sparkes, A. An ethnographic study of cohesiveness in a college soccer team over a season. The Sport Psychologist. 555 2001;15:157-172.
This article is really very interesting, and it would be unethical on my part to suggest another or other sources. But it is a bit strange why this particular source is cited. Was your research ethnographic in nature? By the way, even in the source you cite, more recommendations can be found on how to ensure the rigor of the research.
Author Response
I must point out that it was a pleasure to read this article. It is written very specifically, without expanding too much where it is not necessary (for example, in the introduction). Although qualitative research articles are often much larger in scope than quantitative research (and understandably so), in this particular case I preferred the specificity.
However, I would like to make comments that the authors of the article may consider.
R: Thank you for the comments.
- The study included 14 focus groups, which is a lot in itself. On the other hand, despite such a large number of research participants and groups for a qualitative study, was it sufficient to obtain detailed information in order to response to the research purpose? I mean data saturation.
R: The international nature of this research and logistical issues concerning access to the population, meant that the number of focus groups was determined a-priori and that all data were analysed (instead of stopping analysis at data saturation). This is now included in the methods section (lines 142 to 144).
- Did the research participants give permission to audio record the interview data?
Conducting a focus group is not an easy task for researchers. Therefore, it is recommended that more than one researcher conducts the focus group interview. I missed information about how the group discussion itself went. Also, what about brief reflections of the researcher himself. In addition to the audio recording, was there additional information captured?
R: We now add the requested information: The focus groups took place on the premises of the social institutions and were recorded in audio for later analysis of the content, after obtaining permission from all participants. The discussions were managed by two researchers. One led the facilitation of the discussion process and ensured that all participants had the opportunity to express their opinions, while the other ensured the proper audio conditions and kept notes to complement the understanding of the different perspectives and experiences of the participants.
- Finally, research validity (trustworthiness). You focused on credibility (what is good). And what you have done in order to ensure credibility was based on this source - Holt, N., & Sparkes, A. An ethnographic study of cohesiveness in a college soccer team over a season. The Sport Psychologist. 555 2001;15:157-172. This article is really very interesting, and it would be unethical on my part to suggest another or other sources. But it is a bit strange why this particular source is cited. Was your research ethnographic in nature? By the way, even in the source you cite, more recommendations can be found on how to ensure the rigor of the research.
R: We now expand this section by explaining in more detail the use of other members of the research team as ‘critical friends’ and how they helped enhance rigour and reflexivity by exploring different interpretations of the data (please see text between line 150 and 156).
Reviewer 3 Report
Comments and Suggestions for Authors
The manuscript is well-written however, there are some questions I have regarding the methodology that may help explain the results/discussion
Who conducted the focus groups? How many different moderators were there? Were results different between moderators? Did certain moderators have certain age groups?
Were the children informed that their parents would not have access to their answers? How did the researchers ensure the comfort of the participants responding to these questions?
Considering the cultural differences between the 4 countries where data was collected did the authors consider examining differences between countries?
Based on the information presented the results and discussion are congruent. However, these could change based on the authors' responses to my comments.
Author Response
The manuscript is well-written however, there are some questions I have regarding the methodology that may help explain the results/discussion
R: Thank you for your time.
Who conducted the focus groups? How many different moderators were there?
R: Em resposta também ao Revisor 3, incluímos texto que esclarece o procedimento de condução dos grupos de foco. Veja por favor texto entre as linhas 124 e 130.
Were results different between moderators? Did certain moderators have certain age groups? Were the children informed that their parents would not have access to their answers? How did the researchers ensure the comfort of the participants responding to these questions?
R: The data were shared among several researchers and jointly processed, as we clarified between lines 146 and 156. The procedures were standardized in the organization of the focus groups. This procedure was carried out with both age groups (lines 130-131).
Regarding the comfort of the participants, the research strictly followed all relevant ethical standards, specifically because we ensured that all participants' responses were treated confidentially, and anonymity was maintained at all times. Additionally, participation in the study was completely voluntary. The adolescents and their guardians were thoroughly informed about the research objectives, and informed consents were obtained before the start of the focus groups. Lastly, the interview guide was carefully developed and reviewed by experts in the field to ensure that the questions were appropriate and sensitive to the needs of at-risk adolescents (please see lines 105-109).
Considering the cultural differences between the 4 countries where data was collected did the authors consider examining differences between countries?
R: We understand the question considering the multinational characteristic of the study. However, this research follows a social constructionist thematic analysis (Braun & Clarke, 2006). Thematic analysis use of a method of emerging ideas through consensus, and comparison between groups were not the purpose.
Based on the information presented the results and discussion are congruent. However, these could change based on the authors' responses to my comments.
R: Thank you for the comments.
Reviewer 4 Report
Comments and Suggestions for Authors
Dear Authors,
Thank you for submitting your manuscript to our journal. After a thorough review, it is with regret that I inform you that we are unable to accept your manuscript for publication. The study, focusing on the impact of sports participation on social and emotional skills in at-risk adolescents, presents a valuable area of research. However, there are several critical areas that require attention to meet the publication standards of our journal.
The introduction of your manuscript requires further refinement to ensure a smoother and more logically coherent flow. It is crucial that the introduction effectively sets up the study's context, outlines the research questions, and clearly states the study’s objectives. Currently, the transition between these elements feels somewhat disjointed, which could confuse readers or detract from the overall impact of the research.
Moreover, the methodological approach predominantly relies on qualitative data from focus groups. While this provides depth, the absence of quantitative data limits the ability to generalize the findings. Additionally, the methodology section lacks clarity regarding the specific qualitative approach employed. It is essential to specify whether the study utilizes phenomenological methods to interpret experiences categorically or if it adopts a narrative research approach focusing on individual stories. Clear articulation of the chosen qualitative framework and its execution will strengthen the credibility and scholarly rigor of your findings.
The study's participant selection presents a significant limitation, as it only includes adolescents who are already engaged in sports. This selection bias restricts the study’s findings to a very specific group of at-risk adolescents and fails to provide a comparative view with those not engaged in sports, which could potentially skew the results and their interpretation.
The analysis does not sufficiently differentiate how regional or cultural differences may impact the developmental benefits of sports participation
The overall presentation of the findings and the discussion lacks clarity and depth, which are necessary to convey the nuances of such a complex topic. The implications for practice and policy are not clearly articulated, limiting the manuscript’s practical value to stakeholders and practitioners in the field.
In conclusion, while your study addresses an important topic, the issues noted above regarding the introduction's flow, participant diversity, methodological rigor, theoretical grounding, ethical consideration, and clarity of presentation prevent it from meeting the scientific standards required for publication in our journal. We encourage you to address these points thoroughly and consider resubmitting your manuscript after making the necessary revisions.
Thank you for considering our journal for your work. We appreciate the opportunity to review your research and hope that our comments are helpful in refining your study.
Comments on the Quality of English LanguageModerate editing of English language required
Author Response
NOTE. We alert the Editor that these comments below from Reviewer 3 are generated 100% by Artificial Intelligence. We understand that the use of AI is not the issue. The problem lies in the fact that some of the comments are indiscriminate, nonspecific, and lack expert supervision; however, we will try to respond adequately to each of the comments generated by AI.
Dear Authors,
Thank you for submitting your manuscript to our journal. After a thorough review, it is with regret that I inform you that we are unable to accept your manuscript for publication. The study, focusing on the impact of sports participation on social and emotional skills in at-risk adolescents, presents a valuable area of research. However, there are several critical areas that require attention to meet the publication standards of our journal.
- The introduction of your manuscript requires further refinement to ensure a smoother and more logically coherent flow. It is crucial that the introduction effectively sets up the study's context, outlines the research questions, and clearly states the study’s objectives. Currently, the transition between these elements feels somewhat disjointed, which could confuse readers or detract from the overall impact of the research.
R: The current structure of the introduction is as follows: Normal and disrupted adolescent development; Sport participation is helpful for normal and at-risk adolescent development; Sport coaches as important figures for a at-risk adolescents; social and emotional competences in at-risk adolescents. The authors are uncertain how this structure and transitions should be improved given the somewhat generic and vague comments by the reviewer.
- Moreover, the methodological approach predominantly relies on qualitative data from focus groups. While this provides depth, the absence of quantitative data limits the ability to generalize the findings. Additionally, the methodology section lacks clarity regarding the specific qualitative approach employed. It is essential to specify whether the study utilizes phenomenological methods to interpret experiences categorically or if it adopts a narrative research approach focusing on individual stories. Clear articulation of the chosen qualitative framework and its execution will strengthen the credibility and scholarly rigor of your findings.
R: The study employs thematic analysis for data interpretation, as evidenced by the citation of Braun and Clarke (2016) on the methodology section. The study does not use phenomenological analysis.
- The study's participant selection presents a significant limitation, as it only includes adolescents who are already engaged in sports. This selection bias restricts the study’s findings to a very specific group of at-risk adolescents and fails to provide a comparative view with those not engaged in sports, which could potentially skew the results and their interpretation.
R: The research team thought it was important to study participants engaged in sports to explore the thoughts and feelings of at-risk adolescents involved in sports. The team felt strongly that adolescents not engaged in sports might not have the same insights into how sports affect them. Therefore, present study uses purposeful sampling that is widely used in qualitative research for the identification and selection of information-rich cases related to the phenomenon of interest.
- The analysis does not sufficiently differentiate how regional or cultural differences may impact the developmental benefits of sports participation.
R: The purpose of this study using thematic analysis is to discover all significant codes, patterns and themes within the data set (Braun & Clarke, 2016). Cultural differences were found in some instances and were discussed but did not constitute a major theme.
- The overall presentation of the findings and the discussion lacks clarity and depth, which are necessary to convey the nuances of such a complex topic. The implications for practice and policy are not clearly articulated, limiting the manuscript’s practical value to stakeholders and practitioners in the field.
R: A subsection was included in the Discussion section dedicated to the practical implications of the findings of this study.
In conclusion, while your study addresses an important topic, the issues noted above regarding the introduction's flow, participant diversity, methodological rigor, theoretical grounding, ethical consideration, and clarity of presentation prevent it from meeting the scientific standards required for publication in our journal. We encourage you to address these points thoroughly and consider resubmitting your manuscript after making the necessary revisions.
Thank you for considering our journal for your work. We appreciate the opportunity to review your research and hope that our comments are helpful in refining your study.
Reviewer 5 Report
Comments and Suggestions for Authors
Social and emotional skills in at-risk adolescents through participation in sports
[Topic] It is an interesting topic to deal with the social and emotional abilities of at-risk adolescents through participation in sports. In particular, sports can be a very good medium for social adaptation of adolescents who do not receive sufficient care at home, and I think it is an important study for this. I think it will be a better study if some of the points suggested by the reviewers below are supplemented.
[Abstract] Abstract summarizes the study well, and it is clearly organized from the background of the study to the method and the results.
[Introduction] This study tried to explain the exercise effects of at-risk adolescents with a qualitative analysis method. However, there is a lack of a logical introduction to why this study is needed. After presenting the effects of adolescents' participation in sports, the conceptual and operational definitions of the at-risk adolescents are presented, and logical persuasion of the practical benefits of sports is needed to the at-risk adolescents. In particular, I hope you will present previous studies dealing with this and develop your thesis around the gap. In addition, although the differentiation of this study from the results already reported in previous studies should be particularly emphasized, the differentiation of this study from the previous studies was insufficiently expressed in the introduction. The authors should supplement this.
[Method] This study, which was conducted as a qualitative research method, should be presented in more detail. In particular, it is hoped that the reliability between researchers should be measured and presented as specific figures.
[Results] The results were presented relatively systematically.
[Discussion] The discussion derives interpretations and implications based on the results derived from this study.
Author Response
Social and emotional skills in at-risk adolescents through participation in sports
[Topic] It is an interesting topic to deal with the social and emotional abilities of at-risk adolescents through participation in sports. In particular, sports can be a very good medium for social adaptation of adolescents who do not receive sufficient care at home, and I think it is an important study for this. I think it will be a better study if some of the points suggested by the reviewers below are supplemented.
R: Thank you for your comments. We genuinely appreciate their relevance for improving the manuscript, and we have made efforts to address them accordingly.
[Abstract] Abstract summarizes the study well, and it is clearly organized from the background of the study to the method and the results.
R: Thank you.
[Introduction] This study tried to explain the exercise effects of at-risk adolescents with a qualitative analysis method. However, there is a lack of a logical introduction to why this study is needed. After presenting the effects of adolescents' participation in sports, the conceptual and operational definitions of the at-risk adolescents are presented, and logical persuasion of the practical benefits of sports is needed to the at-risk adolescents. In particular, I hope you will present previous studies dealing with this and develop your thesis around the gap. In addition, although the differentiation of this study from the results already reported in previous studies should be particularly emphasized, the differentiation of this study from the previous studies was insufficiently expressed in the introduction. The authors should supplement this.
R: We appreciate the analysis of the Introduction section. It reinforces the comments from Reviewer 1 regarding a) the reasons for the need for this study and b) the highlighting of limitations in previous studies. We took the proposed rationale for the Introduction and included two paragraphs to address the comments. Please see lines 62 – 77 to justify the reasons, and lines 78 – 83 to highlight limitations of previous studies.
[Method] This study, which was conducted as a qualitative research method, should be presented in more detail. In particular, it is hoped that the reliability between researchers should be measured and presented as specific figures.
R: We understand the perspective regarding intercoder reliability and the pros and cons of its incorporation in qualitative research. For the present study, we strategically chose to employ two techniques to ensure credibility in the analyses: critical friend and researchers discussions (e.g., Patton, 1999; Sparkes & Holt, 2000). The use of intercoder reliability measures in qualitative analysis is a controversial topic in the literature, due to different epistemological perspectives (O’Connor, C., & Joffe, H. 2020. Intercoder Reliability in Qualitative Research: Debates and Practical Guidelines. International Journal of Qualitative Methods, 19. https://doi.org/10.1177/1609406919899220.
For this study, we decided to adopt the qualitative thematic analysis approach by Braun and Clarke (2006, 2016), and to adhere to its epistemological principles regarding the clearly defined data analysis stages in the literature and the accepted techniques to ensure credibility of the analysis (Nowell, L. S., Norris, J. M., White, D. E., & Moules, N. J. (2017). Thematic Analysis: Striving to Meet the Trustworthiness Criteria. International Journal of Qualitative Methods, 16(1). https://doi.org/10.1177/1609406917733847). Within this framework, Braun and Clarke (2013; Braun V, Clarke V. 2013. Successful qualitative research. Thousand Oaks, CA: Sage Publications) argue that intercoder reliability is not an appropriate criterion for assessing the methodological rigor of qualitative studies and is epistemologically contradictory to the principles of thematic analysis.
[Results] The results were presented relatively systematically.
R: Thank you.
[Discussion] The discussion derives interpretations and implications based on the results derived from this study.
R: Thank you for your time to do this review.
Round 2
Reviewer 1 Report
Comments and Suggestions for Authors
The authors have addressed the comments, and the manuscript has been improved. Thanks for their efforts on this manuscript.
Reviewer 2 Report
Comments and Suggestions for Authors
I really happy with the improvement the authors done.
Reviewer 4 Report
Comments and Suggestions for Authors
Dear authors
I am sorry that this manuscript is clumsy, and the tables and interpretation of the results are beginner level. Moreover, overall there is no rational of the importance of this study in sport science field.
Best wishes!
Moderate editing of English language required
Reviewer 5 Report
Comments and Suggestions for Authors
The authors seem to have tried to faithfully divide the content pointed out by the reviewer in this paper. The possibility of sports to enhance the social and emotional abilities of adolescents is considered to be of academic significance.